# Effect of Rotation and Hole Arrangement in Cold Bridge-Type Impingement Cooling Systems†

**Lorenzo Cocchi** *, **Alessio Picchi** and **Bruno Facchini**

Department of Industrial Engineering, Università degli Studi di Firenze, via Santa Marta 3, 50139 Firenze, Italy; alessio.picchi@htc.unifi.it (A.P.); bruno.facchini@unifi.it (B.F.)

* Correspondence: lorenzo.cocchi@htc.unifi.it; Tel.: +39-055-2758-713
† This paper is an extended version of our paper in Proceedings of the European Turbomachinery Conference ETC13, Lausanne, Switzerland, 8–12 April 2019; Paper No. 310

**Abstract:** Experimental activity has been performed to study different impingement cooling schemes in static and rotating conditions. Geometry replicates a leading-edge cold bridge system, including a radial supply channel and five rows of film-cooling and showerhead holes. Two impingement geometries have been studied, with different numbers of holes and diameters but with equal overall passage area. Reynolds numbers up to 13,800 and rotation numbers up to 0.002 have been investigated (based on an equivalent slot width). Tests have been performed using a novel implementation of transient heat transfer technique, which allows correct replication of the sign of buoyancy forces by flowing ambient temperature air into a preheated test article. Results show that complex interactions occur between the different features of the system, with a particularly strong effect of jet supply condition. Rotation further interacts with these phenomena, generally leading to a slight decrease in heat transfer.

**Keywords:** Gas turbine cooling; leading edge; heat transfer; impingement; rotation

---

## 1. Introduction

Current performance levels of gas turbine engines, both in terms of thermal cycle efficiency and specific power output, have been reached thanks to an increase in turbine inlet temperature. The downside of this trend is the intensification of thermal loads on the engine components, which require the development of more effective cooling systems to be managed. The airfoil leading-edge (LE) is one of the most critical regions from this point of view, since it houses the hot gases stagnation point. Moreover, due to the surface curvature, the outer side has a significantly higher extension than the inner side, on which cooling systems can be applied. In order to maximize heat pickup in this region, a common cooling scheme consists of a series of coolant jets, which are generated by a radial supply duct and impinge on the LE inner side [1].

When coolant jets impinge onto a concave surface, a strongly different flow field and heat transfer pattern is induced compared to a flat plate, since with a sharp curvature target the jets can impinge mainly on the side walls of the surface [2]. Target surface curvature also intensifies the effects of the arrangement of impingement holes in terms of heat transfer pattern shape, even if with minor effects in average terms [3].

Film-cooling (FC) and showerhead (SH) holes, where the cooling flow exits the blade, are mainly positive for impingement heat transfer, since they prevent the development of a detrimental crossflow inside the LE cavity [4]. However, hole location can significantly influence heat transfer in either a positive or a negative way [5], while flow distribution between the various extraction rows seems to have mainly local effects [6,7].

Jet supply condition is also relevant for system performance. In cooling systems where the impingement plate is fed by a cavity with a radial crossflow, a residual component of radial momentum can be transported inside the impingement hole and then into the LE cavity: the cooling jet is thus bent towards the radial direction, reducing the heat transfer [8,9]. However, the reduction of the apparent hole cross-section can enhance jet lateral spreading, thus increasing wall-jet interaction and compensating heat transfer degradation [10,11].

If jet impingement is employed to cool a blade LE, rotation influences the internal flow field and heat transfer pattern too. A common outcome of many recent works [12,13] is that Coriolis forces can bend the impingement jets and thus reduce the heat transfer performance. Coriolis forces also act on the flow inside the radial supply channel, which can lead to an uneven supply condition for the impingement jets [11].

All of these studies demonstrate that rotational effects are difficult to predict, since rotation acts on the different and interacting features of the system: consequently, the whole cooling device needs to be replicated in order to perform a reliable investigation. Starting from this consideration, in this work a complete cold bridge-type geometry has been experimentally studied, with the aim to determine rotational effects on the cooling performance of the system. The effect of impingement hole number and diameter has also been determined by comparing two impingement geometries with a different hole pattern but with the same overall passage area.

## 2. Geometry and Test Conditions

The investigated geometry (Figure 1) replicates a scaled up leading-edge cooling system. A radial feeding channel provides coolant air to an array of square-edged round holes, which generate a series of jets impinging on the inner surface of the LE cavity. In this work, two different impingement geometries have been compared, denoted as GTE-B and GTE-C respectively, whose features are reported in Table 1. GTE-C presents twice the number of impingement holes of GTE-B and a smaller hole diameter, but in order to perform a significant comparison the two geometries have the same overall geometric passage area: this allows investigation of the effects of a different coolant distribution with similar average jet velocities and coolant consumption. The value for the passage area has been defined considering a realistic ratio with the LE section, while the staggered arrangement has been selected since shifting holes towards the LE sides is expected to provide higher heat transfer for a wedge-shaped cavity [7]. For both geometries, hole axis is inclined $35.6°$ towards the pressure side.

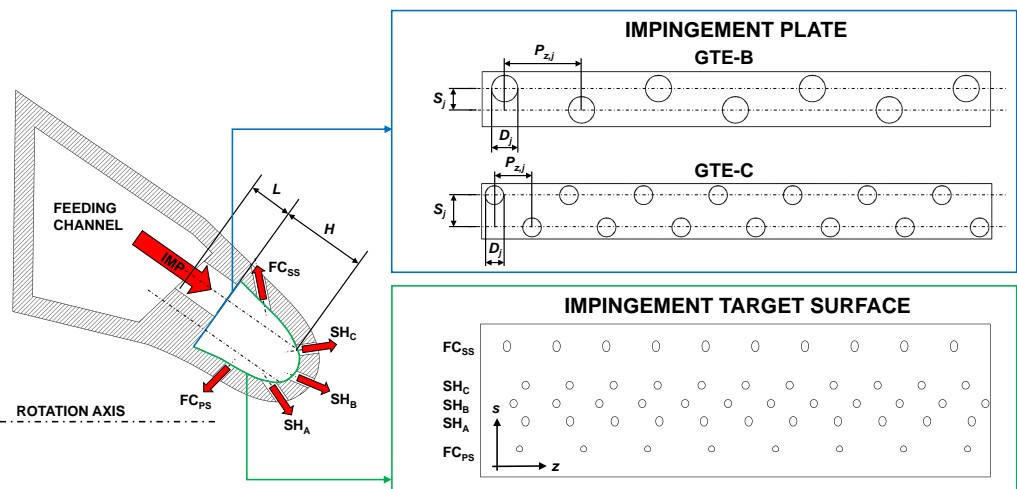

**Figure 1.** Scheme of the cooling geometries.

**Table 1.** Investigated systems geometric characteristics.

| Geometry | | GTE-B | | GTE-C | |
|---|---|---|---|---|---|
| | | **PS Row** | **SS Row** | **PS Row** | **SS Row** |
| Jet number | $N_j$ [-] | 3 | 4 | 7 | 7 |
| Jet diameter | $D_j$ [mm] | 20.0 | 20.0 | 14.14 | 14.14 |
| Radial spacing | $P_{z,j}$ [mm] | 58.6 | 58.6 | 28.3 | 28.3 |
| Tangential spacing | $S_j$ [mm] | | 16.0 | | 24.0 |
| Jet-to-target surface distance | $H$ [mm] | 48.1 | 45.7 | 43.3 | 39.5 |

Coolant flow is then discharged by five rows of extraction holes; the three central ones represent showerhead holes and the side ones film-cooling holes. All extraction holes are 5.6 mm in diameter, with the exception of pressure side FC holes which are 4.8 mm. The mass flow rate through each of the extraction rows has been set to replicate the effect of the external pressure distribution: FC row located on the pressure side (PS) draws 10% of the total impingement mass flow rate $\dot{m}_{IMP}$, FC row on the suction side (SS) draws 30% of $\dot{m}_{IMP}$ and the three SH rows 20% of $\dot{m}_{IMP}$ each. A part of the coolant flow radially entering the supply channel does not feed the impingement holes and leaves the geometry at its radial outer extremity. By varying the radially oriented mass flow rate in the feeding channel (i.e., crossflow) it is possible to set different upstream conditions for the impingement jets, corresponding to different sections of the blade. In particular, the hub, midspan and tip sections have been investigated by setting the mass flow rate leaving the feeding channel equal to 70%, 40% and 10% of the total coolant amount entering the blade. The ratio between these last two mass flow rates will be defined as crossflow ratio ($Cr$) and will be employed to identify the three sections.

Since the two geometries share the same overall passage area, the equivalent width of a 2D slot $b$ has been chosen as the characteristic length for data reduction, given by

$$b = \frac{A_{IMP}}{Z_{max}} = \frac{N_j \frac{\pi D_j}{4}}{Z_{max}} \tag{1}$$

where $A_{IMP}$ is the overall passage area and $Z_{max}$ is the total radial extension of the heat transfer surface; $b$ is the same for the two geometries and is equal to 5.68 mm.

Test conditions have been set in terms of Reynolds number, defined as

$$Re_b = \frac{\dot{m}_{IMP} b}{A_{IMP} \mu} \tag{2}$$

where $\mu$ is air dynamic viscosity. Four equally spaced $Re_b$ values ranging from 5400 to 13,800 have been investigated, corresponding to impingement mass flow rates ranging approximately between 40 g/s and 100 g/s. For every $Re_b$ value tests have been performed in static and rotating conditions: for the latter ones, the similitude in terms of rotational effects is obtained by replicating the rotation number, defined as

$$Ro_b = \frac{\omega b \rho A_{IMP}}{\dot{m}_{IMP}} \tag{3}$$

where $\omega$ is the rotational speed and $\rho$ is air density. All the rotating tests have been performed at a constant $Ro_b$ value of 0.002. Heat transfer results will be presented as Nusselt number

$$Nu_b = \frac{hb}{k} \tag{4}$$

where $h$ is convective heat transfer coefficient and $k$ is air thermal conductivity evaluated at thermochromic liquid crystals (TLC) event temperature.

## 3. Test Rig

### 3.1. Measurement Apparatus

Heat transfer measurements have been performed thanks to a dedicated test rig, a scheme of which is reported in Figure 2. The rig consists of an open loop suction type wind tunnel installed on a rotating chassis. A rotary joint supports the chassis, performs the pneumatic connection and allows data transmission and power supply to the onboard instrumentation thanks to a series of slip rings. Rotation is obtained thanks to a 7.5 kW electric motor, and air circulation is allowed by four vacuum pumps with 2400 m$^3$/h total capacity.

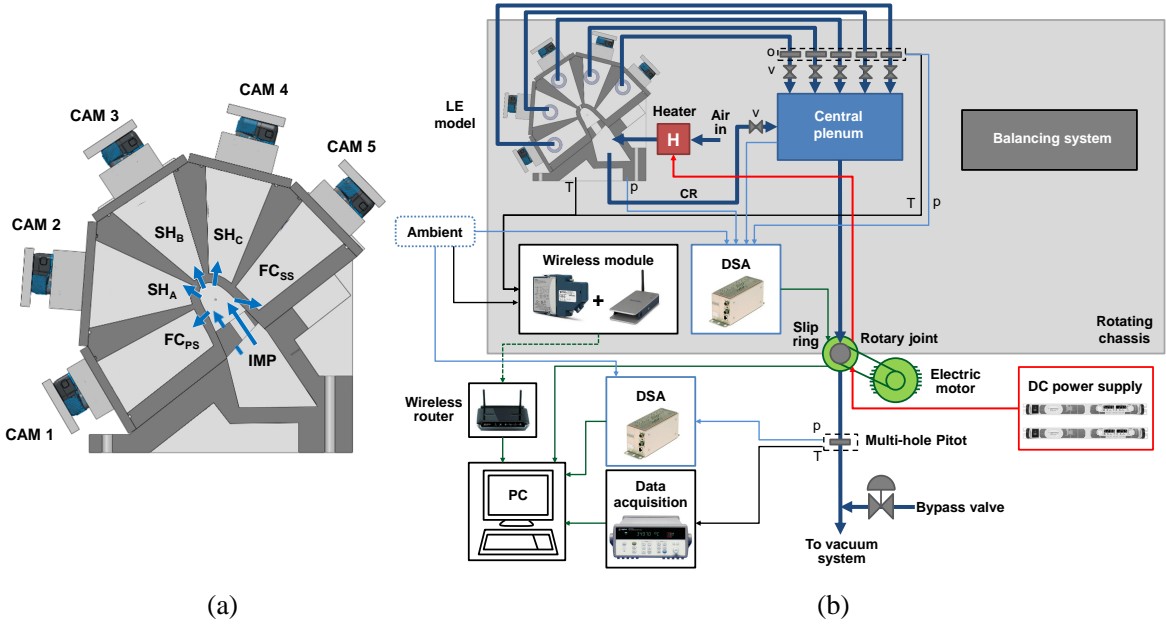

(a)                                                                    (b)

**Figure 2.** (**a**) Cross-section of the cooling geometry and (**b**) scheme of the test rig.

Heat transfer measurements have been performed with the transient heat transfer technique using TLCs: to provide the quick and uniform temperature step required by such method, air enters the rig passing through a 2 kW mesh heater. Air then feeds the model of the cooling geometry at its inner radial extremity. In order to obtain both optical access and thermal insulation for the inner surfaces, the model is entirely made of transparent PMMA. The flow passing through the impingement holes is extracted from the geometry through the five extraction rows, each connected to an independent plenum directly built on the outer side of the LE. This design avoids radial mass flow unbalances along the extraction rows in rotating conditions, since the same centrifugal pressure gradients are present inside the LE cavity and the plenums. Independent mass flow set up and measurement is obtained for each extraction row thanks to a gate valve and a calibrated orifice installed on the extraction line of each plenum. The five extraction lines and the feeding channel outlet are all connected to a central plenum, from which the total mass flow rate is extracted through the rotary joint. The pneumatic line connecting the joint to the vacuum system houses a multi-hole Pitot (for overall mass flow rate measurement) and a fast actuation bypass valve (needed by the measurement technique which will be described in the next section).

Static pressure measurement is performed by two arrays of piezoresistive relative sensors with 16 channels each, allowing mass flow rate measurement and rig flow management. Air temperature acquisition is performed by T type thermocouples: in particular, an unsheathed thermocouple is located on the centerline of each impingement hole, allowing accurate temperature history measurement

and independent data reduction for each jet. Temperature uniformity at the beginning of the test is assessed thanks to various thermocouples embedded into the LE surface. Temperature evolution of the heat transfer surface is obtained thanks to a coating of narrow band TLCs (Hallcrest R40C1W), with a color play range from 40 °C to 41 °C. TLCs have been calibrated using the steady state gradient method [14] in the same optical conditions of the actual rig. Moreover, before the calibration TLCs undergo similar thermal cycles as the actual tests to take into account thermal hysteresis effects. TLC response is recorded thanks to five GoPro HERO5 Black cameras, installed on the test article outer walls and set to record a 1920 × 1080 pixel video stream at 29.97 fps.

*3.2. Measurement Technique*

The replication of the correct sign of buoyancy forces in rotating conditions requires the heat transfer surfaces to be at a higher temperature than the coolant air. To achieve this goal, a novel implementation of the TLC transient technique has been developed, which has been presented and validated in a previous work by the authors [15] and will be briefly described in the following. Before the test, the correct flow conditions are set by flowing the whole nominal mass flow rate inside the rig at ambient temperature. Once this is done, the bypass valve located on the rig outlet line is opened, so that the mass flow rate passing through the rig decreases due to its pressure losses. In the meantime the heater is switched on, heating up the inlet air flow and thus the test article itself. The heating process is carried on until the model reaches a uniform and constant temperature higher than the TLC event (usually 55–57 °C), which takes around 90 min. The uniformity of the model temperature is obtained thanks to the test article design, since air at the same temperature is present on both sides of the jet target surface (see Figure 2). At the beginning of the test, a trigger simultaneously shuts down the heater and closes the bypass valve: in this way, the whole nominal mass flow rate at ambient temperature enters the hot model, thus obtaining the desired cold temperature step.

The camera frames recorded during the test are processed to extract the time in which each pixel reaches the maximum intensity of the green color, i.e., the so-called event temperature (obtained through the TLC calibration). Such correspondence between time and temperature allows the transient conduction inside the solid to be solved at every point on the surface [16], obtaining a local value of convective heat transfer coefficient $h$. In order to increase the accuracy, the actual air temperature history has been considered using the superimposition principle [17]. Performing this operation for each camera provides five $h$ distributions, which are combined using a custom 3D mapping procedure.

Measurement accuracy has been calculated according to the standard ANSI/ASME PTC 19.1 [18] based on Kline and McClintock [19] method. Maximum uncertainty on Reynolds number is 16% for the $Re_b$ = 5400 tests, while for the other tests, values range from 8% (for $Re_b$ = 8200 test) to 4% (for $Re_b$ = 13,800 test). Maximum uncertainty on Nusselt number is around 12% in local terms and 10% in average terms, with typical values around 9.5%. All the uncertainties are reported with a level of confidence of 95%.

## 4. Results

In Figure 3 Nusselt number distributions on the LE inner surface for $Cr$ = 40%, $Ro_b$ = 0 tests with different $Re_b$ values are reported for the investigated geometries. The heat transfer surface is unrolled on a flat plane, where the horizontal coordinate $S$ represents the distance along the surface itself from its extremity close to the PS, while the vertical coordinate $Z$ is along the outward radial direction. Both coordinates are scaled with respect to their maximum values. Cross symbols mark the intersection between jet axes and heat transfer surface, while the locations of the extraction holes are shown by solid lines. Only the part of the surface not suffering from boundary effects is reported, corresponding to an area including the impact locations of jets 3-7 for GTE-B and of jets 5-14 for GTE-C (progressively numbering the jets from the test hub to the tip of the test article).

Considering GTE-B results, a high heat transfer region is present for each jet, generally oriented towards the lateral side of the target surface closer to the corresponding impingement

hole. Such regions are elongated in the horizontal direction, which may be attributed to the surface curvature leading the jet shear layer to interact with the side walls: in fact, local maxima are often present on the sides rather than in the central part of the LE (e.g., at $Z/Z_{max}$= 0.35, 0.5, 0.65, 0.8). The heat transfer pattern is also strongly influenced by the presence and location of extraction holes, as shown by the clear distortion of the $Nu_b$ distribution. Moreover, secondary peaks surrounding such holes are present. Since the extraction holes patterns relative to each jet are different, strongly different $Nu_b$ distributions are present from jet to jet.

As far as GTE-C is considered, a single peak is still present for each jet, strongly concentrated closer towards the side walls and elongated towards the center. Peaks on the PS look much stronger than the ones on the SS, which leads to the PS jets with 13% higher average heat transfer than SS jets and may be attributed to the supply condition. In fact, the impingement plate upstream side is inclined 11° in the tangential direction with respect to the hole axis, as visible in Figure 1. As a consequence, flow turning at the inlet section is lower on the PS with respect to the SS, and the associated lower pressure losses may actually lead the coolant to flow preferably along this side of the hole. For this geometry, this phenomenon leads to a lower apparent jet-to-target surface distance for the jets closer to the PS with respect to the SS, providing higher flow velocities along the pressure side of the target surface and thus higher heat transfer values. This phenomenon is also expected to occur for GTE-B geometry, but its effects on the heat transfer pattern are likely to be weakened by the strong influence of extraction holes observed in the previous case. The GTE-C heat transfer distribution also presents a more periodic pattern in the radial direction with respect to GTE-B. This fact can be interpreted considering that due to the smaller hole diameter, coolant jets mainly impinge on the side walls of the profile, where a lower number of extraction holes per unit area is present with respect to the central region. As a result, the heat transfer pattern can be expected to be less influenced by the extraction holes in this region, thus providing a regular heat transfer pattern.

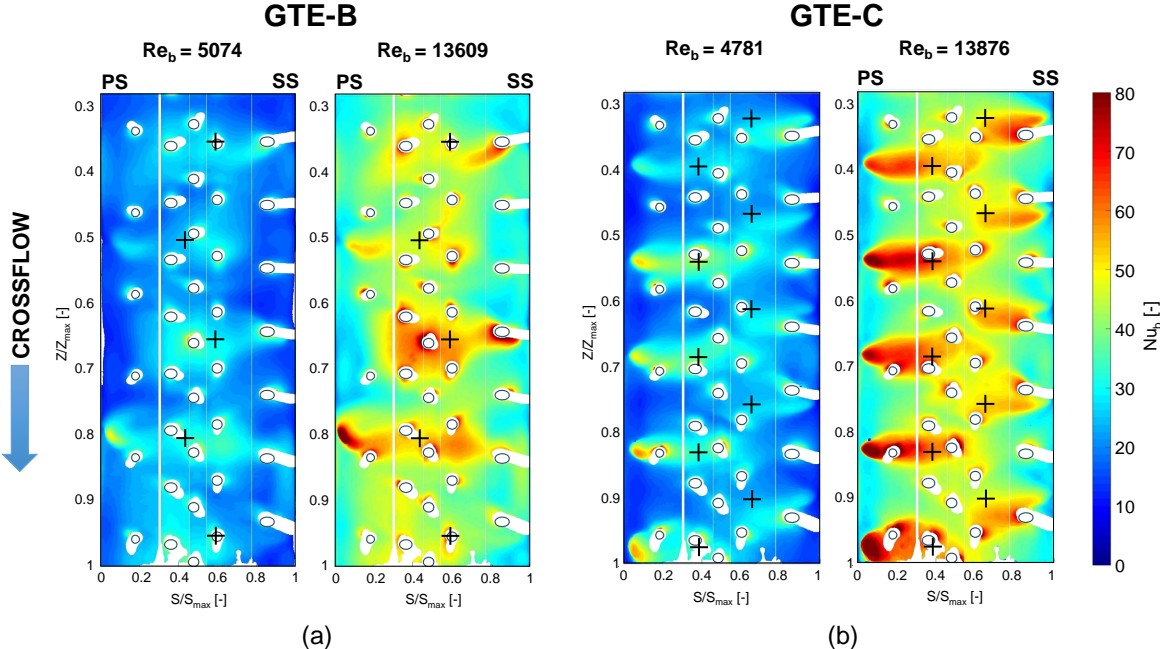

**Figure 3.** $Nu_b$ distributions for Cr = 40%, $Ro_b$ = 0 for (**a**) GTE-B and (**b**) GTE-C geometries.

Despite the phenomena described above, it must be noticed that for both geometries the highest average heat transfer occurs in the central region, which presents an area average $Nu_b$ 25% higher than the side walls for GTE-B and 20% for GTE-C.

Figure 3 also reveals that by increasing Reynolds number heat transfer increases in every location, but without significant alteration of its pattern shape. This effect can be quantified by considering area averaged $Nu_b$ values reported in Figure 4, which appear to be a close function of $Re_b^{0.6}$.

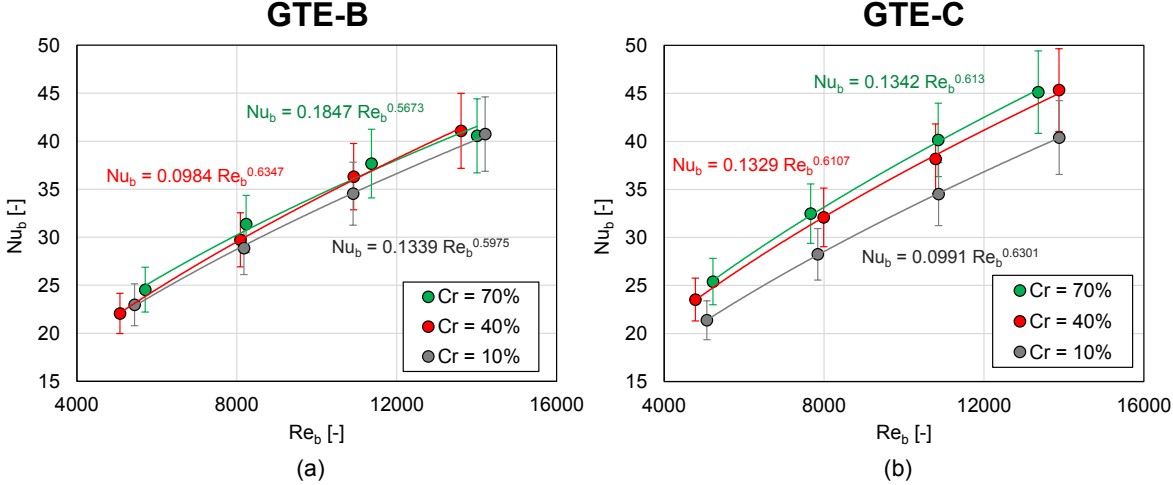

**Figure 4.** $Nu_b$ area averaged values for $Ro_b = 0$ for (**a**) GTE-B and (**b**) GTE-C geometries.

## 4.1. Effect of Crossflow Condition

Figure 5 presents heat transfer distributions obtained for different crossflow conditions, i.e., for different radial sections of the blade, for both the investigated geometries in static and rotating conditions. Since slightly different $Re_b$ values were recorded in the various tests, in this case $Nu_b$ values have been scaled with respect to $Re_b^{0.6}$ to remove this effect from the data. The heat transfer maps show that a higher crossflow leads to an increase of $Nu_b$ peaks magnitude and extension in every condition. This also causes an increase in area averaged heat transfer, which can also be appreciated by considering Figure 4 reported in the previous section. In particular, GTE-B seems to be less sensible to crossflow effects than GTE-C: passing from Cr = 10% to Cr = 70%, an average heat transfer increase of 4.6% is recorded for GTE-B and of 14.1% for GTE-C. Sensitivity to crossflow also seems to increase in rotating conditions, since area averaged $Nu_b$ increases of 9.9% for GTE-B and of 18.4% for GTE-C. The observed phenomena can be interpreted by considering the effect of upstream crossflow on jet shape. As shown by previous investigations [11], inside the impingement holes the residual radial momentum drives the flow towards the downstream side: this reduces the apparent size of the passage section, whose shape becomes more elliptical than circular, and also increases the flow velocity. The elliptical shape is also expected to be maintained by the impingement jets, which thus experience an enhanced lateral spreading and a higher velocity. Since for the present cases heat transfer seems to be mainly influenced by the interaction between jets and side walls, it can be expected that such phenomenon has a positive effect.

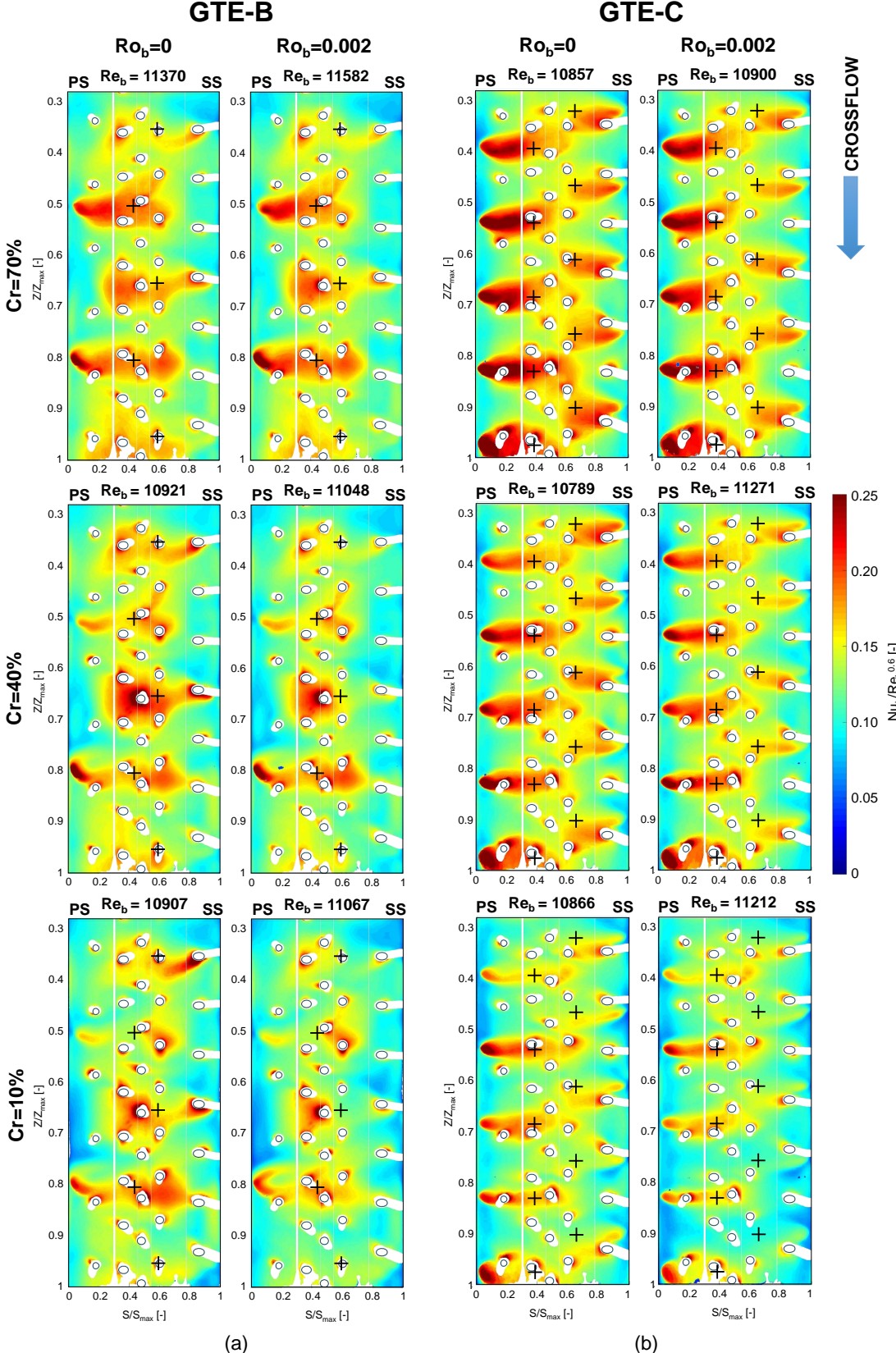

**Figure 5.** $Nu_b/Re_b^{0.6}$ distributions for $Re_b = 11{,}000$ for (**a**) GTE-B and (**b**) GTE-C geometries.

*4.2. Effect of Rotation*

By observing the $Nu_b$ distributions of Figure 5, it can be noticed that in rotating conditions heat transfer decreases on the side walls ($S/S_{max} < 0.3$ and $S/S_{max} > 0.7$) for both geometries. Such decrease is almost negligible at Cr = 70% and maximum at Cr = 10%. This phenomenon, which has already been observed in a previous study by the authors [15], can be explained by considering that the impingement holes are directed towards the PS (see Figure 1) and that during rotation Coriolis force with a radial inward direction is generated inside the holes [12]. As a consequence, the Coriolis force may balance a part of the residual crossflow momentum (which has a radially outward direction), causing a reduction of jet lateral spreading and thus of jet-wall interaction. This effect seems to increase as crossflow decreases, i.e., as the magnitudes of Coriolis and crossflow related inertia forces become more similar.

Figure 5 also shows that heat transfer reduction in rotating conditions seems weaker on PS than on SS. This can be justified by considering that inside the supply channel Coriolis forces drive the flow along the PS during rotation, leading total pressure to increase in this region [11]. As a consequence, the coolant supply may be enhanced for the jets closer to the PS, thus weakening the negative rotational effects in this region.

Rotational effects can be quantified by considering area averaged $Nu_b$ values, which are reported in Figure 6. For both GTE-B and GTE-C it can be observed that heat transfer reduction due to rotation decreases as crossflow increases, changing from an average of around −6% at Cr = 10% to −2% at Cr = 70% in the investigated $Re_b$ range. Figure 6 also shows that GTE-B and GTE-C geometries have almost identical heat transfer performance at Cr = 10%, i.e., when the lowest jet lateral spreading is present. As crossflow increases, GTE-C starts providing higher heat transfer than GTE-B in both static and rotating conditions, with an 8% average $Nu_b$ increase at Cr = 40% and a 10% increase at Cr = 70%. As a consequence, it seems that smaller holes allow a better exploitation of the increased jet lateral spreading in high crossflow conditions, which is likely related to the possibility of distributing the coolant closer to the cavity side walls.

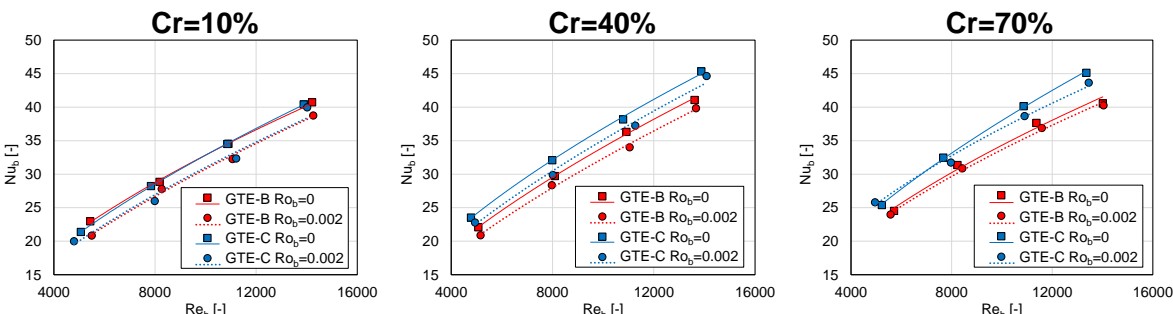

**Figure 6.** $Nu_b$ area averaged values for different crossflow conditions.

*4.3. Analysis of Overall Cooling Performance*

The analysis performed in the previous paragraphs revealed that the highest heat transfer values are generally associated with GTE-C geometry. Even so, in order to determine the actual cooling performance also pressure losses need to be taken into account. In fact, even if the two geometries have the same geometric passage area, the different holes diameter, length, and location may lead to different pressure losses for a given overall mass flow rate. In order to take this phenomenon into account, Figure 7 presents area averaged $Nu_b$ values as a function of the pumping power $W = \dot{m}\Delta p/\rho$, where $\Delta p$ is the pressure drop across the impingement plate. $Nu_b$ and $W$ values are scaled with respect to the corresponding values $Nu_{b,0}$ and $W_0$ of GTE-B static test in minimum $Re_b$ and $Cr$ conditions. Power law fittings are also reported for each dataset. The charts reveal that in every condition GTE-C is associated with a lower pumping power (i.e., lower pressure losses) and provides equal or higher heat transfer than GTE-B. Similar trends are also observed for rotating tests, not reported for the

sake of brevity. As a consequence, it can be stated that the best overall cooling performance in the investigated range of parameters is provided by GTE-C geometry. The superior performance of GTE-C is particularly evident with large *Cr* values, given the stronger heat transfer enhancement obtained as the jet lateral spreading increases.

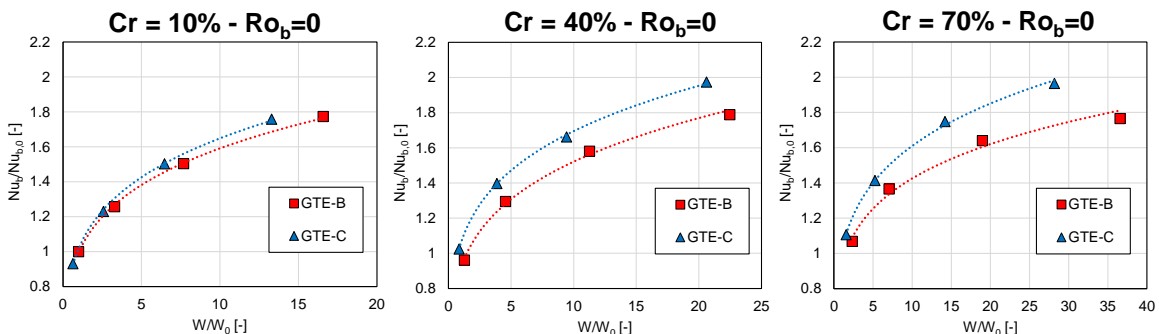

**Figure 7.** $Nu_b$ area averaged values as a function of pumping power for static tests.

## 5. Conclusions

In this work, a cold bridge-type leading-edge cooling system has been experimentally studied, with the aim to determine the rotational effects on heat transfer. The whole cooling system has been replicated, including the radial supply channel and the coolant extraction holes, thus allowing the study of the mutual interaction of the different features. The use of two different impingement geometries, with different number of holes and diameter but equal passage area, also allowed the effect of different coolant arrangements to be studied. The analysis has been performed by exploiting a novel implementation of the transient heat transfer technique using TLCs, which allows the application of such method to a rotating investigation while replicating the correct sign of buoyancy forces. The main findings of the present research are the following:

- Both staggered jet arrays generate horizontally elongated heat transfer peaks on the side walls of the leading-edge cavity. The effect of extraction holes on the heat transfer pattern increases with the number of holes located close to the jet stagnation point.
- Heat transfer increases as crossflow grows, i.e., moving from the tip to the hub of the blade (up to 18%), which has been attributed to the higher lateral spreading and velocity of the jets and thus to the increased interaction with the side walls. This effect increases if the coolant is distributed closer to the side walls.
- Rotation slightly decreases heat transfer and such effect intensifies as crossflow decreases, which has been attributed to the Coriolis effects on the impingement jets. Rotation seems to act also on the feeding channel, unbalancing the heat transfer distribution towards the PS. This also demonstrates that all the features of a cold bridge cooling system need to be taken into account to reliably study rotational effects.
- The two impingement geometries have similar overall cooling performance at low crossflow values. On the other hand, as crossflow grows the geometry housing a bigger number of smaller holes seem to better exploit the increased jet-wall interaction, providing up to 10% higher heat transfer. The better cooling performance is also confirmed if the required pumping power is taken into account.

**Author Contributions:** Conceptualization, L.C., A.P. and B.F.; methodology, L.C., A.P. and B.F.; software, L.C.; validation, L.C., A.P. and B.F.; formal analysis, A.P. and B.F.; investigation, L.C.; resources, A.P. and B.F.; data curation, L.C.; writing–original draft preparation, L.C. and A.P.; writing–review and editing, L.C., A.P. and B.F. visualization, L.C.; supervision, A.P. and B.F.; project administration, B.F.; funding acquisition, B.F.

**Funding:** This research received no external funding. The APC was funded by Euroturbo.

**Conflicts of Interest:** The authors declare no conflict of interest.

## Nomenclature

| | |
|---|---|
| *A* | area [m$^2$] |
| *b* | 2D slot equivalent width [m] |
| *Cr* | crossflow ratio [−] |
| *D* | diameter [m] |
| *H* | jet-to-target surface distance [m] |
| *h* | convective heat transfer coefficient [W/m$^2$ K] |
| *k* | thermal conductivity [W/mK] |
| *L* | hole length [m] |
| $\dot{m}$ | mass flow rate [kg/s] |
| *N* | number [−] |
| *Nu* | Nusselt number [−] |
| *P* | spacing [m] |
| *p* | static pressure [Pa] |
| *Re* | Reynolds number [−] |
| *Ro* | Rotation number [−] |
| *S* | tangential spacing [m] |
| *T* | temperature [K] |
| *W* | pumping power [W] |

**Acronyms**

| | |
|---|---|
| *FC* | Film Cooling |
| *PMMA* | Polymethyl Methacrylate |
| *PS* | Pressure Side |
| *SH* | Showerhead |
| *SS* | Suction Side |
| *TLC* | Thermochromic Liquid Crystals |

**Greeks**

| | |
|---|---|
| *μ* | dynamic viscosity [Pa s] |
| *ρ* | density [kg/m$^3$] |
| *ω* | rotational speed [rad/s] |

**Subscripts**

| | |
|---|---|
| *b* | 2D slot |
| *IMP* | impingement section |
| *j* | impingement jet |
| *max* | maximum |
| *z* | radial direction |
| 0 | GTE-B $Re_b = 5400$, $Cr = 10\%$ static test |

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
