# Peer review of "Effect of Rotation and Hole Arrangement in Cold Bridge-Type Impingement Cooling Systems†"

_ijtpp, doi:10.3390/ijtpp4020013_

Round 1

Reviewer 1 Report

Mandatory Request Changes: Requested changes which are essential for the understanding and completeness of the paper. Paper of author(s) who have not complied with these requests may be rejected.:

None

Recommended Requested Changes: Changes will improve the quality of the paper. Authors are strongly encouraged to comply with these requests.:

In general small spelling mistakes throughout the paper, I recommend a linguistic review.

In the introduction write thermal cycle efficiency instead of just efficiency. On page 3 it's said that the two geometries (B and C) has the same overall passage area, is this referred to the geometric area or the effective flow area? Please clarify. It would be nice to know the pressure loss of respective impingement plate used, or at least the change in pressure loss. A few explaining sentences of the selection of cooling schemes, it seems cumbersome to draw any conclusions other than the back-toback comparison made in the study... is there any general trend by increased hole size or hole distribution. Does it confirm earlier findings? Is there any comparisons that can be made to a numerical model of the setup, this in order to support the argumentation of flow physics

Author Response

Dear reviewers,

First of all, the authors would like to express their gratitude to you for your useful and constructive observations. The paper has been revised according to the provided corrections, and the various suggestions were exploited in order to improve the paper quality. As a consequence, the authors believe that the work is now much more robust and complete.

Apart from the changes implemented in the paper, specific answers to the different comments are also reported in this document, in order to provide the requested information that could not be included in the paper.

Kind regards,

L. Cocchi, A. Picchi, B. Facchini

Reviewer 2 Report

Mandatory Request Changes: Requested changes which are essential for the understanding and completeness of the paper. Paper of author(s) who have not complied with these requests may be rejected.:

See additional file

Recommended Requested Changes: Changes will improve the quality of the paper. Authors are strongly encouraged to comply with these requests.:

See additional file

Author Response

(The authors gave the same response as above.)

Reviewer 3 Report

Mandatory Request Changes: Requested changes which are essential for the understanding and completeness of the paper. Paper of author(s) who have not complied with these requests may be rejected.:

--

Recommended Requested Changes: Changes will improve the quality of the paper. Authors are strongly encouraged to comply with these requests.:

Pg 3, 1st paragraph, last sentence: "inclined of 35.6" -> "inclined 35.6"

Pg 4, 2nd last sentence: "2400 m^3/h" is this normal cubic meters? (Nm^3/h)

Pg5. 2nd paragraph, last sentence: "sesnors"->"sensors"

Fig3. Perhaps mention that crosses mark intersection of the jet axis and the surface.

Pg 7. 1st paragraph, 19th line, "impinge in the side walls" -> "impinge on the side walls"

Author Response

(The authors gave the same response as above.)
